

# Universal performance gap of neural quantum states applied to the Hofstadter-Bose-Hubbard model

Eimantas Ledinauskas[1,2⋆] and Egidijus Anisimovas[1†]

**1** Institute of Theoretical Physics and Astronomy, Vilnius University
Saulėtekio al. 3, LT-10257, Vilnius, Lithuania
**2** Baltic Institute of Advanced Technology, Pilies St. 16-8, LT-01403, Vilnius, Lithuania

⋆ eimantas.ledinauskas@ff.vu.lt , † egidijus.anisimovas@ff.vu.lt

## Abstract

Neural Quantum States (NQS) have demonstrated significant potential in approximating ground states of many-body quantum systems, though their performance can be inconsistent across different models. This study investigates the performance of NQS in approximating the ground state of the Hofstadter-Bose-Hubbard (HBH) model, an interacting boson system on a two-dimensional square lattice with a perpendicular magnetic field. Our results indicate that increasing magnetic flux leads to a substantial increase in energy error, up to three orders of magnitude. Importantly, this decline in NQS performance is consistent across different optimization methods, neural network architectures, and physical model parameters, suggesting a significant challenge intrinsic to the model. Despite investigating potential causes such as wave function phase structure, quantum entanglement, fractional quantum Hall effect, and the variational loss landscape, the precise reasons for this degradation remain elusive. The HBH model thus proves to be an effective testing ground for exploring the capabilities and limitations of NQS. Our study highlights the need for advanced theoretical frameworks to better understand the expressive power of NQS which would allow a systematic development of methods that could potentially overcome these challenges.

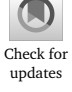

# 1 Introduction

Artificial neural networks have been the driving force behind numerous groundbreaking results in diverse fields, demonstrating their vast potential and adaptability (e.g. [1–5]). This impressive progress has also sparked interest in the application of neural networks within the field of many-particle quantum systems. In the context of ground state approximation, Ref. [6] pioneered the use of neural networks as an ansatz in the variational Monte Carlo setting. Since then, the application of neural quantum states (NQS) to various physical models has achieved notable success (for recent reviews, see Ref. [7–9]). In some scenarios, NQS have significant advantages over other state-of-the-art techniques. For example, unlike matrix product states, NQS can represent states with volume-law entanglement [10–13]. Additionally, NQS are capable of representing ground states of non-stoquastic Hamiltonians, which pose a challenge to quantum Monte Carlo algorithms due to the sign problem [14]. Yet, it was found that for certain physical models under particular parameter conditions NQS performance notably declines. One such example is the Heisenberg $J_1$-$J_2$ model with $J_2/J_1 \approx 0.5$ which results in highly frustrated system [15,16]. Another instance is the Newman–Moore model on a triangular lattice which can exhibit glassy behavior depending on the lattice size [17]. Both Refs. [15] and [17] argue that a rugged variational landscape might be a primary concern, as NQS can become trapped in saddle points or local minima, resulting in a poor representation of the ground state. Studying such intrinsically complex systems and mapping out the precise conditions that make them difficult for NQS is crucial for identifying the limitations of NQS methods, comparing different approaches, and improving them based on the observations obtained.

In this study, we explore the capability of NQS to approximate the ground state of a system of hard-core bosons on a two-dimensional square lattice, which is subject to a uniform magnetic field perpendicular to the lattice. This system is henceforth referred to as the Hofstadter-Bose-Hubbard (HBH) model. Our work is complementary to the recent study [18] that implemented recurrent neural network (RNN) wave functions and mainly focused on long-range interacting systems. Also, a ladder version of the HBH model was explored in Ref. [19]. When the magnetic flux values reach a point where the associated magnetic length is comparable to the lattice period, the two scales can compete, potentially leading to frustration phenomena

in this model. Indeed, we observe that an increase in magnetic flux correlates with a sharp decrease in NQS performance. Remarkably, the energy error can differ by three orders of magnitude between models with low and high magnetic flux values, even in very small system sizes. We also demonstrate that, at a fixed magnetic field strength, there is a straightforward relationship between NQS performance and the ratio of the number of NQS weights to the dimension of the Hilbert space. This relationship holds for both multilayer perceptron (MLP) and convolutional neural network (CNN) architectures with varying depth and width, as well as for different lattice sizes and particle densities. Consequently, the observed performance decline appears to be quite universal, suggesting that the magnetic field introduces features that are fundamentally challenging for such unspecialized NQS architectures. In the limit of finite on-site interactions Ref. [18] also reported a drop in performance, evidenced by the chosen benchmark observables, the energy and the expectation value of the Hubbard interactions term. To understand the reasons behind this decline in NQS performance, we conducted a series of numerical experiments exploring various potential explanations. Interestingly, unlike the findings in Ref. [15, 17], we do not observe any evidence that the issues stem from the ruggedness of the loss landscape. This makes the HBH model even more interesting, as it potentially represents a novel type of challenging system that may require different approaches to achieve high precision. Additionally, we demonstrate that the issues are unlikely to arise directly from complex phase structure, quantum entanglement, or the fractional quantum Hall effect. Based on these findings, we believe that the HBH model serves as a valuable test case for exploring the capabilities and limitations of NQS methods, as well as for developing novel architectures and optimization methods. The decline in NQS performance is not easily resolvable, highlighting the complexity and richness of the problem.

Our paper is organized as follows. Section 2 describes the HBH model. Section 3 provides a brief overview of NQS, the neural network architectures used in this study, and the various NQS optimization methods employed to approximate the ground states. In Section 4, we present the results of our numerical experiments, demonstrating the universal decline in NQS performance with increasing magnetic flux and investigating various potential causes. Finally, we conclude with a brief summarizing Section 5.

## 2 Hofstadter-Bose-Hubbard model

The Hofstadter model [20, 21] was introduced to describe a two-dimensional charged particle moving on a lattice pierced by a uniform perpendicular magnetic field. The interplay of the lattice periodicity with the length scale set by the magnetic field leads to the emergence of a fractal energy spectrum, as two structures – that of the Bloch bands and of the Landau levels – need to be imposed simultaneously. The model supports a topological band structure and a fractal phase diagram for the integer quantum Hall effect [22–26]. Its controllable physical realization is accessible in ultracold atomic experimental setups [27–29]. Thanks to a successful inclusion of particle interactions in an experiment [30], the model was upgraded to the many-body setting, paving the way towards the study of the fractional quantum Hall regime [31–34].

In our work, we focus on the Hofstadter-Bose-Hubbard (HBH) model, i.e. the bosonic version of the interacting Hofstadter model in the limit of infinitely strong contact interactions. Given a square lattice of $L_x \times L_y$ sites, the Hamiltonian reads

$$H = -J \sum_{j=1}^{L_x} \sum_{k=1}^{L_y} \left( a_{j+1,k}^\dagger a_{j,k} + a_{j,k+1}^\dagger a_{j,k} \, e^{i2\pi\alpha j} \right) + \text{h.c.} \tag{1}$$

Here, the lattice sites are indexed by an ordered pair of integer indices $(j, k)$ and $a_{j,k}^\dagger$ $(a_{j,k})$ are the standard creation (annihilation) operators obeying bosonic commutation rules. Transitions in the second ($y$) direction feature the Peierls phase [35] $2\pi\alpha j$ whose magnitude is linearly proportional to the first ($x$) coordinate $j$. This implements the Landau-gauge description of a uniform magnetic field, and moving around each square plaquette counter-clockwise a particle accumulates the net phase $2\pi\alpha$ corresponding to the scaled artificial flux $\alpha$. In this model, the flux $\alpha$ enters in a periodic fashion; taking into account also the symmetry of the energy spectrum with respect to the sign inversion $\alpha \to -\alpha$, it is sufficient to study the interval $\alpha \in (0, 0.5)$. Note that due to the hardcore constraint – no more than one particle per site – interactions do not feature explicitly in the Hamiltonian (1). Instead, their role is enacted by limiting the set of the accessible states. The studied Hilbert space is spanned by the computational basis $|s\rangle$ including all possible configurations of a given number $N$ of particles on $L_x \times L_y$ lattice sites such that all sites are either empty or singly occupied. Starting from the vacuum (empty lattice) state $|\text{vac}\rangle$, the basis states are built as

$$|s\rangle := a_{j_1,k_1}^\dagger a_{j_2,k_2}^\dagger \cdots a_{j_N,k_N}^\dagger |\text{vac}\rangle, \tag{2}$$

such that all the index pairs $(j, k)$ are distinct and ordered according to a fixed prescription. The states in the computational basis $|s\rangle$ are then also ordered and can be indexed by a single index $s$ running from 1 to the maximum state index

$$s_{\text{max}} = \frac{(L_x L_y + N - 1)!}{N!\,(L_x L_y - 1)!}. \tag{3}$$

Let us emphasise that we study the model with open boundary conditions in order to be able to vary the scaled flux $\alpha$ continuously. By the same token, our model is lacking any translational or rotational symmetries. In our work, we mainly focus on small lattices and assume that any long-ranged interactions (that would couple to neighbours) are negligible. We chose these simplifications because such model is already challenging for NQS with large $\alpha$ values and the absence of other possibly problematic details helps to better isolate the causes. Also, smaller simpler models require less computational resources to simulate and thus allow richer analysis.

## 3 Methods

### 3.1 Neural quantum states

In this work, we examine many-particle quantum systems with finite degrees of freedom. The quantum wave function for such systems is represented by a complex vector as follows:

$$|\psi\rangle = \sum_s \psi(s)|s\rangle, \tag{4}$$

where $|s\rangle$ denotes vectors of a computational basis, and $\psi(s) = \langle s|\psi\rangle$ represents the corresponding elements of the wave function vector. Typically, the dimension of the wave function vector grows exponentially with the size of the system (e.g., the number of sites in the lattice or the number of particles).

A neural quantum state (NQS) [6] is defined as a neural network that takes a basis vector $|s\rangle$ as input and outputs the corresponding wave function element $\psi(s)$. The motivation behind NQS is to leverage neural networks to discern and utilize the structure in the mapping between $|s\rangle$ and $\psi(s)$. This approach aims to approximate the wave function using fewer variational parameters than the dimension of the full wave function. Consequently, it enables the modeling of larger systems than what exact methods feasibly allow.

## 3.2 Neural network architecture

In this work, to implement neural quantum states, we employ both MLP and CNN architectures. The input configuration $s$ is encoded as either a vector (in the case of MLP) or a matrix (in the case of CNN), representing site occupations with ones or zeros based on the hard-core boson assumption. We also symmetrize the input values to be either -1 or 1, instead of 0 and 1. In the case where the neural quantum state (NQS) weights are real, the final layer in both architectures is a densely connected layer. This layer maps the outputs of the last hidden layer to two numbers, interpreted as the real and imaginary parts of $\log \psi(s)$. When dealing with complex weights, the output is a single complex number that directly represents $\log \psi(s)$. In the case of the CNN, the output from the last hidden layer is flattened into a vector. This step is taken to break the translational invariance, as the system under consideration does not exhibit this symmetry. After each densely connected or convolutional layer, we apply layer normalization [36] followed by the GELU activation function [37]. We also experimented with various different input/output encoding, normalization, and activation functions used in the literature but found that these details do not significantly affect the main results of this work.

Unless stated otherwise, we consistently limit the size of the neural networks to ensure that the number of variational parameters is significantly less than the dimensionality of the problem's Hilbert space, even when working with small lattices. This constraint is intentional because employing NQS with a higher dimensionality than the state vectors to be approximated would undermine the utility of using NQS in the first place. In cases where the network's dimensionality exceeds that of the state vectors, those vectors could be optimized directly, negating the benefits of employing NQS.

## 3.3 Ground state approximation with NQS

For approximating the ground state with NQS we employ three distinct methods, which are detailed in the subsections that follow. These methods are described briefly here; for a more comprehensive overview, please refer to Ref. [8, 9, 38–40].

### 3.3.1 Stochastic reconfiguration

The first method minimizes the average energy using stochastic reconfiguration (SR) [41, 42]. The update rule for the parameters in this method can be described as follows:

$$\theta_{n+1} = \theta_n - \gamma S^{-1} \partial_\theta E, \tag{5}$$

where $\theta_n$ denotes the parameter vector at the $n$-th iteration step and $\gamma$ denotes a learning rate multiplier. Here, $S = O^\dagger O$ represents the quantum geometric tensor, with $O_s = \partial_\theta \log \psi_{\text{NQS}}(s)$ being the vector of wave function derivatives, and $\partial_\theta E$ is the gradient of the average energy. This gradient is computed according to:

$$\partial_\theta E = \langle H_{\text{loc}} O^* \rangle - \langle H_{\text{loc}} \rangle \langle O^* \rangle, \tag{6}$$

where $H_{\text{loc}}$ stands for the local Hamiltonian estimator, defined by:

$$H_{\text{loc}}(s) = \sum_{s'} \frac{\psi(s')}{\psi(s)} \langle s|\hat{H}|s' \rangle. \tag{7}$$

The advantage of this method over straightforward gradient descent lies in its exploitation of the quantum geometric tensor, which grounds the optimization process within the geometric structure of the quantum state space [43].

To implement this method, we utilize the NETKET Python [44] package [45, 46]. We utilized the ADAM optimizer [47] with a fixed learning rate of $3 \cdot 10^{-4}$ and a batch size of 512. We added a small diagonal shift of 0.01 to the quantum geometric tensor.

### 3.3.2 Supervised imaginary time evolution

The second method we employ simulates imaginary time evolution via the supervised learning of a target wave function [48–50]. This approach computes the target wave function from the current NQS wave function by applying a small Euler method time step, $\Delta\tau$, in accordance with the Schrödinger equation with imaginary time. The calculation of the target wavefunction is as follows:

$$\psi_\text{T}(s) = \psi_\text{NQS}(s) - \Delta\tau \sum_{s'} \psi_\text{NQS}(s')\langle s|\hat{H}|s'\rangle \,. \tag{8}$$

The supervised learning process involves maximizing the overlap between $\psi_\text{NQS}(s)$ and $\psi_\text{T}(s)$, achieved by minimizing the loss function:

$$L = -\log\left(\frac{|\langle\psi_\text{NQS}|\psi_\text{T}\rangle|^2}{\langle\psi_\text{NQS}|\psi_\text{NQS}\rangle\langle\psi_\text{T}|\psi_\text{T}\rangle}\right)\,. \tag{9}$$

This loss function is minimized through gradient descent, with the gradient given by:

$$\partial_\theta L = 2\Re\left(\langle O^*\rangle - \left\langle\frac{\psi_\text{T}(s)}{\psi_\text{NQS}(s)}\right\rangle^{-1}\left\langle\frac{\psi_\text{T}(s)}{\psi_\text{NQS}(s)}O^*\right\rangle\right)\,. \tag{10}$$

The target wave function is held fixed for multiple steps and is updated after energy decreases below a certain threshold value. The time step $\Delta\tau$ is chosen to minimize the average energy of the target wave function. Throughout this paper, we refer to the latter method as supervised imaginary time evolution (SITE). Unlike SR, this method does not require computing the quantum geometric tensor, as the geometric structure of the quantum state space is inherently embedded in the loss function. Another key difference is that in SITE, the target wave function remains fixed for multiple steps, whereas in methods that directly minimize the energy, the effective target wave function changes with each step [50].

For SITE we use our own Python code, developed using JAX [51], OPTAX [52], and FLAX [53] packages. We used the same optimization hyperparameters as those used with SR, specifically a learning rate of $3\cdot10^{-4}$ and a batch size of 512.

### 3.3.3 Supervised learning of exact solution

The third method involves supervised learning directly applied to the ground state wave function $\psi_\text{ED}(s)$ obtained through exact diagonalization (ED). This is achieved by minimizing the loss as specified in Eq. 9, but with $\psi_\text{ED}(s)$ replacing $\psi_\text{T}(s)$. This approach is primarily useful for testing and evaluating the performance of NQS, as having the ED solution available in practice would render the NQS solution unnecessary. Learning the wave function directly in this manner is generally insightful as it assesses whether the expressivity of neural networks is adequate for representing the relevant wave function. This approach is particularly valuable when investigating the SITE method, as it is expected to provide an upper bound for the precision of the ground state approximation. Ideally, performing imaginary time evolution as described in Eq. 8 should eventually yield the target wave function $\psi_\text{T}(s) = \psi_\text{ED}(s)$, resulting in the same loss function as used here. We maintained the same optimization hyperparameters as those used in the SITE method. In the supervised learning scenario described in this section, we assess performance by measuring the overlap deviation from 1, which is defined as:

$$d(\psi_\text{NQS}, \psi_\text{ED}) = 1 - |\langle\psi_\text{NQS}|\psi_\text{ED}\rangle|\,. \tag{11}$$

## 3.4 State sampling

All three methods necessitate the sampling of states, $s$, to estimate the average values appearing in Eq. 6 and 10. Typically, this is accomplished approximately by employing the Metropolis–Hastings algorithm [54, 55] or exactly through the use of autoregressive neural network architectures [56, 57]. However, in this work, we concentrate on system sizes for which exact diagonalization (ED) is feasible. In such cases, it is possible to compute the full NQS wave function vector and thus use it directly for exact sampling. We adopt this approach throughout our work to streamline the training procedures and eliminate complications arising from sampling errors, which would otherwise render the analysis more complex.

# 4 Results

## 4.1 Drop of NQS performance in strong magnetic flux regime

We begin by examining the dependence of NQS performance on the strength of magnetic flux, utilizing SR and SITE for optimization. For this we conduct experiments with a $4 \times 5$ lattice containing 4 particles and employ MLP architecture with two hidden layers, each comprising 32 neurons. This neural network has 1,856 variational parameters, which is more than two times fewer than the dimension of the Hilbert space for this case, which amounts to 4,845. We observed a rapid decline in the energy accuracy of NQS with increasing values of $\alpha$. Figure 1 illustrates the relationship between energy error (obtained by comparing with exact diagonalization) and $\alpha$. The energy error associated with NQS escalates by several orders of magnitude within the $0 < \alpha < 0.2$ range and experiences a further increase of an order of magnitude within the $0.2 < \alpha < 0.3$ range. Importantly, the observed decline in performance is virtually identical whether SR or SITE is used for NQS optimization. This uniformity implies that the underlying issue is unlikely to originate from the optimization algorithm itself.

We also incorporated into Figure 1 the results obtained using the density matrix renormalization group (DMRG) method for comparison. For this we utilised TENPY [58] Python package. We note that this is not the best result that could be achieved with DMRG as the virtual bond dimension of the Matrix Product State (MPS) was set to closely align the number of variational parameters (2,336) with that of our NQS. Notably, NQS shows significantly better performance at lower $\alpha$ values, becoming comparable to DMRG at higher values. While the performance of DMRG also deteriorates with increasing $\alpha$, this decline is considerably less pronounced than in the case of NQS. The employed virtual bond dimension of 8 is relatively small compared to the values typically used for DMRG. Therefore, we also included data with the bond dimension increased to 24. The performance improves significantly and surpasses NQS in the large magnetic flux regime. However, we note that in this case, the MPS contains 20,832 parameters, exceeding the Hilbert space dimension by about four times. Thus, even though the MPS has enough parameters to represent the ground state exactly, the DMRG algorithm fails to find such a solution, highlighting the well-known weakness of DMRG in approximating strongly entangled systems in more than one spatial dimension.

We also observe a similar decrease in accuracy with increasing magnetic flux when directly maximizing the overlap between the NQS and the wave function obtained from ED (described in sec. 3.3.3). The overlap deviation from the target wave function vector increases by orders of magnitude in the regime of strong magnetic flux, compared to the scenario without a magnetic field. Remarkably, the specifics of the neural network architecture, as well as the size and density of the physical model, have minimal impact on this result. To validate this observation, we undertook an extensive series of experiments across a variety of neural network architectures, lattice dimensions, and particle counts. We explored every combination of the following

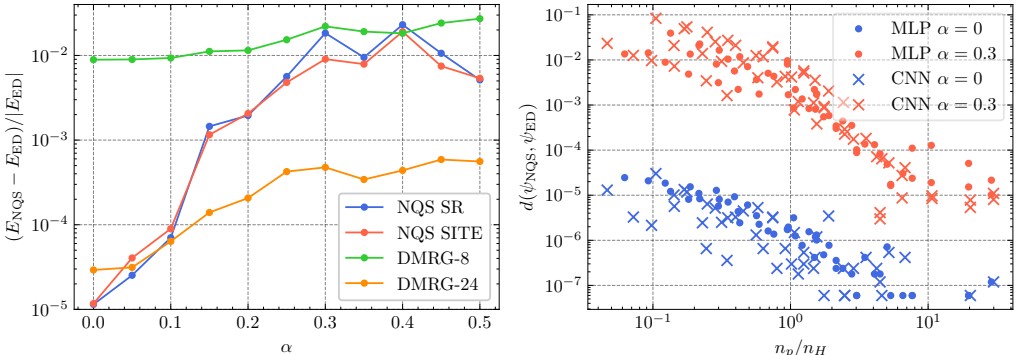

Figure 1: **Left:** Dependence of NQS and DMRG energy error on magnetic flux. The green curve represents DMRG with 8 virtual bonds (a similar number of variational parameters to NQS), while the orange curve corresponds to 24 virtual bonds (approximately 5 times more variational parameters than the Hilbert space dimension). **Right:** Overlap deviation plotted against the ratio of NQS parameters to the dimension of the Hilbert space. Each point represents a different run, varying in neural network architecture, lattice size, and particle number. Blue and red symbols indicate runs without magnetic fields and with strong magnetic fields, respectively. Circles and crosses denote runs using MLP and CNN neural network architectures, respectively.

parameters: the number of layers (2, 3, or 4); the number of neurons per hidden layer (32, 64, or 128); the number of lattice columns (4, 6, or 8); and the number of particles (4, 6, or 8). Additionally, we conducted analogous experiments using a CNN architecture, where we varied the number of channels (10, 20, or 44) instead of the number of neurons. The results from these experiments are illustrated in Fig. 1. Clearly, a relationship exists between the overlap deviation and the ratio of NQS parameters to the number of dimensions in the Hilbert space. The relationship is characterized by a significant dispersion of approximately one order of magnitude. This variability arises because different neural network configurations may vary in their optimality for this problem, and different physical system configurations may present varying levels of difficulty for approximation. Despite this large variation, a consistent overlap deviation gap of three to four orders of magnitude is observed between cases with $\alpha = 0$ and $\alpha = 0.3$.

The observed universality indicates that the decline in accuracy within the strong magnetic flux regime can be efficiently studied by focusing on a single set of parameters, as the insights gained should be transferable across different configurations. Bearing this in mind, our subsequent numerical experiments will focus on a specific parameter set. Specifically, we will utilize a $4 \times 5$ lattice with 4 particles to examine the performance differences between two $\alpha$ values: 0 and 0.3. Furthermore, we will only use MLP neural network architecture, which consists of two hidden layers, each equipped with 32 neurons. Additionally, our focus will shift towards directly learning the wave function obtained from ED through overlap maximization, rather than relying on SR and SITE methods. By streamlining the process in this manner, we not only remove numerous complicating factors but also significantly reduce the computational requirements of our numerical experiments, thereby accelerating the pace of iteration.

There is no general, well-established criterion for determining when the accuracy achieved by a numerical method is sufficient for the problem to be considered solved. However, it is worthwhile to investigate how the number of NQS parameters required to reach a certain accuracy depends on the value of the magnetic flux. For this purpose, we define the minimal required energy error as $\Delta E_{\text{req}} = 0.1 \cdot (E_{\text{exc}} - E_{\text{gs}})$, where $E_{\text{exc}}$ is the energy of the first excited

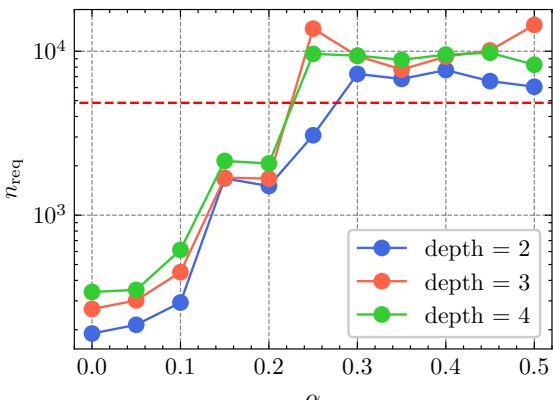

Figure 2: Dependence of the required number of NQS parameters to achieve energy error $\Delta E_{\mathrm{req}}$ (10% of the energy gap between the ground and first excited states) as a function of magnetic flux. The blue, red, and green curves correspond to neural networks with depths of two, three, and four, respectively. The horizontal dashed line indicates the number of dimensions in the Hilbert space.

state and $E_{\mathrm{gs}}$ is the energy of the ground state. Achieving this level of accuracy should ensure that the main features of the ground state are well approximated, and that the resulting state is clearly distinguishable from the excited states. Fig. 2 shows the number of NQS parameters, $n_{\mathrm{req}}$, required to achieve the specified accuracy as a function of magnetic flux $\alpha$. These experiments were conducted by fixing the depth of the MLP neural network to two, three, and four, while varying only the width. The exact values of $n_{\mathrm{req}}$ were obtained through linear interpolation. The number of required parameters increases by several orders of magnitude as $\alpha$ increases, and once again, the observed relationships are not sensitive to the network architecture (in this case, neural network depth). Interestingly, there are sudden jumps in $n_{\mathrm{req}}$ at $\alpha \approx 0.1$ and $\alpha \approx 0.2$, possibly indicating a transition after which the system becomes more difficult for NQS to approximate. Notably, when $\alpha > 0.2$, the required number of parameters significantly exceeds the dimension of the Hilbert space.

## 4.2 Assessment of potential causes for the performance drop

### 4.2.1 Statistics of wave function elements

To identify the potential differences in ground state vectors that may account for the different levels of NQS performance, we analyzed the statistics of wave function elements at $\alpha = 0$ and $\alpha = 0.3$. Figure 3 presents comparison between histograms of the exact wave function vectors at both $\alpha$ values, as well as comparisons between the NQS and exact vectors at the same $\alpha$ values. A notable difference between the ground states at $\alpha = 0$ and $\alpha = 0.3$ is the broader spread of element values at $\alpha = 0.3$. However, this wider distribution should not pose a problem for NQS, as indicated by the rightmost histograms, which show its capability to encompass the required range. We also attempted to approximate a modified $\alpha = 0$ ground state vector such that its values would spread across a similar range (achievable in this case by squaring the elements). This modification resulted in an overlap deviation almost identical to that of the original $\alpha = 0.3$ ground state, thereby dismissing the concern over a broad range of element values as a potential issue. Concerning the histogram of NQS elements at $\alpha = 0.3$, it is evident that it lacks elements within the value range of (-3.2, -2.2) and possesses an excess of values below -3.2. The cause of this discrepancy remains unclear.

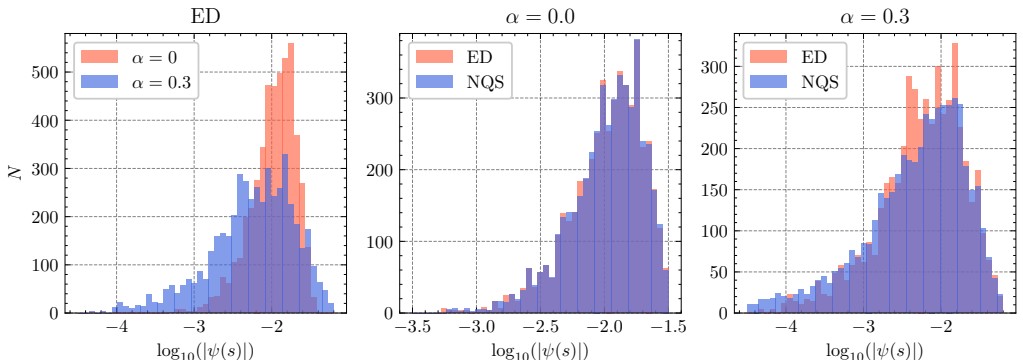

Figure 3: Histograms of wave function element values. **Left:** Comparison of wave functions obtained by ED with (blue) and without (red) a magnetic field. **Middle:** Comparison of NQS (blue) and ED (red) wave functions without a magnetic field. **Right:** Comparison of NQS (blue) and ED (red) wave functions with a strong magnetic field. Areas of overlap are colored purple.

Figure 4 illustrates how the errors in the norm and phase of NQS wave function elements vary in relation to the norm of the corresponding element. Given that equivalent wave functions may differ by a global phase shift, we aligned the global phase by shifting it by $\Delta\phi = \arg(\langle\psi_{\text{NQS}}|\psi_{\text{ED}}\rangle)$ before computing the phase error. There is a notable anticorrelation between phase error and norm at $\alpha = 0.3$. This correlation is anticipated since elements with a higher norm are sampled more frequently during training, thereby exerting a greater influence on the loss. However, no such distinct correlation between the norm of an element and its error is observed at either $\alpha = 0$ or $\alpha = 0.3$.

### 4.2.2 Phase structure

A significant change introduced by the magnetic field is the transition of the Hamiltonian to nonstoquastic regime (the ground state cannot be represented by a wave function with only real positive values). Consequently, the ground state may develop a complicated phase structure, challenging for NQS to learn. This complexity can be analyzed by approximating either the norm or phase with NQS, while deriving the other component from ED. The results of such experiments are shown in Fig. 5, illustrating the dependency of overlap deviation on $\alpha$ when NQS learns the full wave function or only norm/phase of its elements. It is evident that performance deteriorates at nearly the same rate even when a neural network approximates only the norm, which is equivalent to learning a ground state of a stoquastic Hamiltonian. This observation indicates that the phase structure is not the primary challenge. However, it could still be a contributing factor, as learning solely the phase also becomes more difficult at higher $\alpha$ values.

### 4.2.3 Quantum entanglement

Another potentially relevant distinction between the ground states at $\alpha = 0$ and $\alpha = 0.3$ relates to the level of quantum entanglement. To measure entanglement, we divide the system into two parts, perform a Schmidt decomposition on the resulting bipartite system, and then calculate the von Neumann entropy:

$$S = -\sum_k \sigma_k \log_2 \sigma_k \,, \tag{12}$$

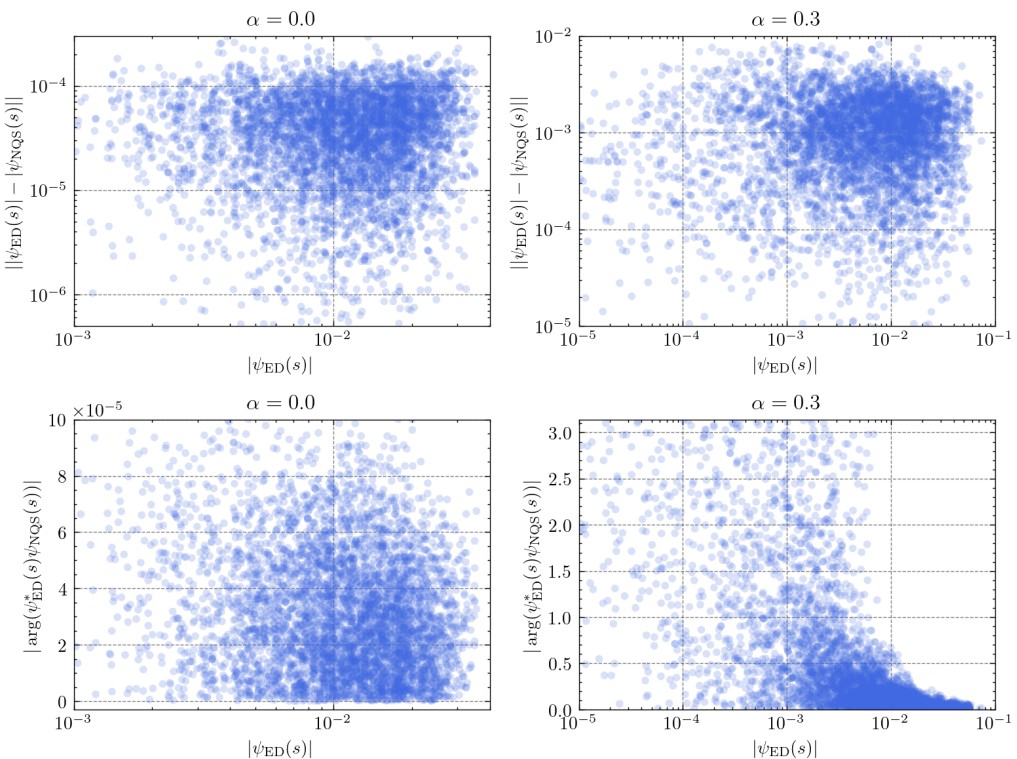

Figure 4: **Upper row:** Errors in the norm of NQS wave function elements plotted against the norm of the corresponding ED wave function elements. **Lower row:** Errors in the phase of NQS wave function elements plotted against the norm of the corresponding ED wave function elements. Both cases shown without (left column) and with a strong (right column) magnetic field.

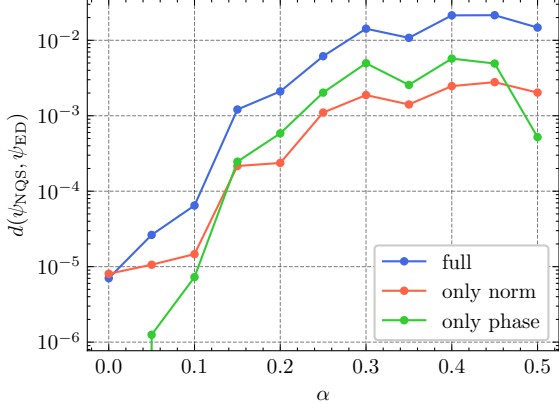

Figure 5: Dependence of overlap deviation on magnetic flux when NQS approximates the full wave function (blue), and only the phase (green) or norm of its elements.

where $\sigma_k$ denotes the singular values obtained from the Schmidt decomposition. The ground state at $\alpha = 0$ exhibits a von Neumann entropy of 1.83, whereas the ground state at $\alpha = 0.3$ has an entropy of 2.5.

To investigate whether the increased entanglement complicates the learning process for NQS, we attempted to artificially reduce the entanglement of the $\alpha = 0.3$ ground state and then learn it with NQS. The entanglement reduction was executed in two distinct manners. The first approach involved truncating the Schmidt decomposition, used in the entropy calcu-

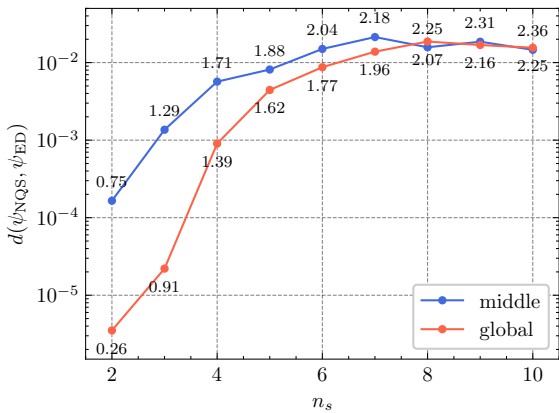

Figure 6: Dependence of overlap deviation on the number of singular values kept after truncating the Schmidt decomposition at the middle lattice point (blue) or at every edge of a 1D flattened lattice (red). Numbers near the points indicate the level of entanglement, measured by von Neumann entropy.

lation, to include only the $n_s$ largest singular values. The second method entailed ordering the lattice sites in a one-dimensional sequence and performing the truncation multiple times at each connection. Figure 6 demonstrates how the overlap deviation varies with the truncation number $n_s$ and the resulting entropy for both methods of entropy reduction. It can be seen that lowering the entropy does result in better overlap. However, at entropy values around 1.8, the overlap deviation is significantly higher compared to the case of $\alpha = 0$, which exhibits the same entropy level. Therefore, it appears that increased entanglement is not the primary factor complicating the learning process for NQS.

### 4.2.4 Fractional quantum Hall effect

As it is well known [22], the noninteracting version of Hofstadter model is a paradigmatic model featuring the emergence of topological single-particle energy bands. In the presence of strong particle interactions and in the regime of a partially filled lowest Landau band, the Hofstadter Bose-Hubbard model has been conjectured to support the fractional Chern insulating (FCI) states, analogous to the Laughlin states in continuous systems [33, 34, 59, 60]. For bosons, the principal FCI state is ecpected at the filling factor of $\nu = 1/2$ particles per flux quantum. Note that in finite systems with open boundaries, the number of plaquettes pierced by the flux is $(L_x - 1) \times (L_y - 1)$, thus the filling factor is defined [61] as $\nu = N/[\alpha(L_x - 1)(L_y - 1)]$, with $N$ denoting the number of particles. The formation of a fractionalized state on matter would certainly pose an additional (and unexplored) challenge for the NQS approach. However, we argue that in our case the performance drop cannot be ascribed to the stabilization of FCI states. In Figure 7 we plot the overlap deviation on the magnetic flux $\alpha$ (left panel) and the corresponding dependence on the filling factor $\nu$ (right panel) for a selection of lattice geometries. We see that the characteristic exponential growth of the overlap deviation is mainly associated with the regime of filling factors well above unity. The behavior of the overlap deviation below $\nu = 1$ is rather erratic and the emerging peaks do not consistently point to a particular value of $\nu$, as one would expect looking for signals of FCI stabilization. All in all, we believe that in our experiments the considered regimes are not favourable for the emergence of FCI states, and they cannot be identified as a source of the performance gap.

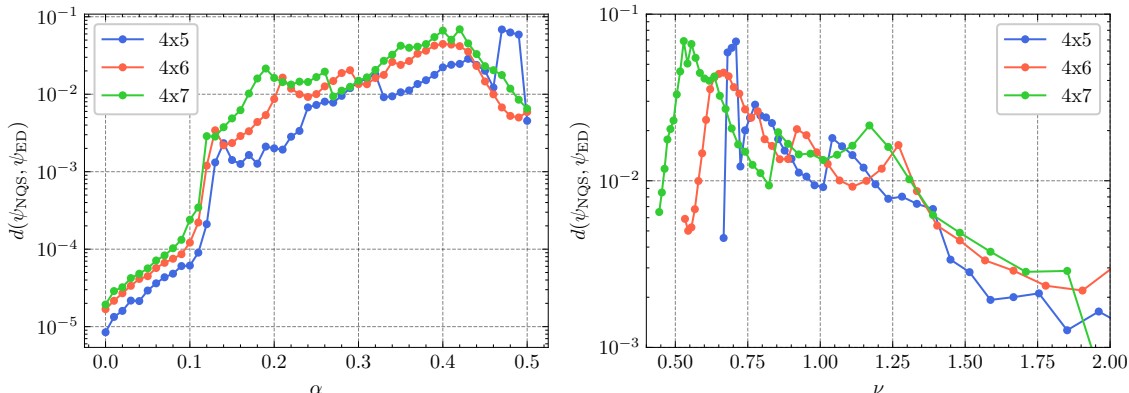

Figure 7: Dependence of overlap deviation on magnetic flux **(left)** and filling factor **(right)** for three different shapes of the lattice: $4 \times 5$ (blue), $4 \times 6$ (red), and $4 \times 7$ (green).

### 4.2.5 Variational loss landscape

As demonstrated in Ref. [15] and [17] with other physical models, the ruggedness of the variational loss landscape may significantly impact the performance of NQS. Following Ref. [17], we apply a loss landscape visualization technique from Ref. [62]. In this method, we select a central point in the NQS parameter space, denoted by $\theta$, and two random unit direction vectors, $e_1$ and $e_2$. We can then visualize the loss landscape as a two-dimensional surface defined by:

$$f(\delta_1, \delta_2) = L(\theta + \delta_1 e_1 + \delta_2 e_2), \tag{13}$$

where $L$ is given by Eq. 9. Unlike during neural network training, we compute the loss using the full NQS wave function vector, ensuring that the loss value is exact and free from errors due to stochasticity. This approach is feasible because we are studying a small physical system. The dimensionality reduction in this technique is dramatic and inevitably results in loss of information. However, such qualitative analysis can still be valuable in practice and has provided various insights into deep neural network learning and generalization [62]. The resulting loss surface for our case with $\alpha = 0.3$ after NQS optimization is depicted in Fig. 8. Interestingly, the landscape appears highly convex and does not exhibit the rugged features identified in Ref. [15,17]. We also inspected visualizations using different sets of random direction vectors $e_1$ and $e_2$, as well as different $\alpha$ values. In all instances, the resulting surfaces were qualitatively similar and showed no signs of ruggedness.

To analyze the variational landscape more quantitatively, we computed the Hessian matrix of the loss function with respect to the NQS parameters after optimization. Typically, Hessian analysis is not practically feasible with neural networks due to the large number of weights involved. However, in our case, the baseline network contains 1,856 weights, making such analysis possible. The spectra of Hessian eigenvalues for different $\alpha$ values are depicted in the left part of Fig. 9. All the eigenvalues are positive, confirming that the extremum identified is a minimum and the local loss landscape is convex. However, the magnitude of the eigenvalues, which reflects the steepness of the minimum, varies significantly with $\alpha$. This variation is illustrated in the right part of Fig. 9, where the dependence of the median eigenvalue on $\alpha$ is plotted. We observe that from $\alpha = 0$ to $\alpha = 0.3$, the steepness increases exponentially. A higher curvature around the minimum could result in a more challenging optimization process, as it becomes more difficult to precisely converge on the minimum. To test whether this prevents convergence at high $\alpha$ values and thus explains the performance decline, we attempted to refine the final neural network weights using Newton's method. Given the local convexity of

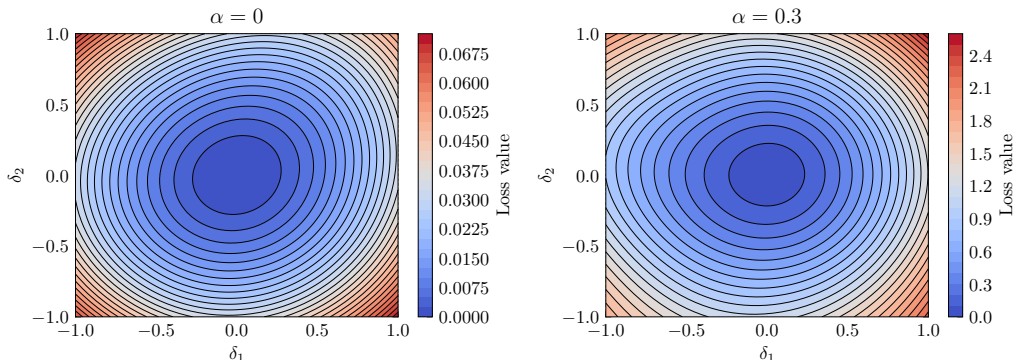

Figure 8: Two-dimensional visualization of the loss landscape near the point of final NQS parameters with $\alpha = 0$ **(left)** and $\alpha = 0.3$ **(right)**.

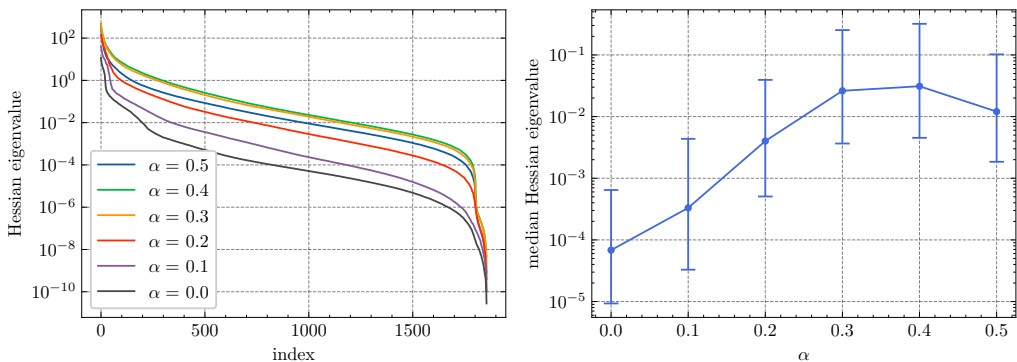

Figure 9: **Left:** Spectra of Hessian eigenvalues, ordered by magnitude, for various magnetic flux values. **Right:** Dependence of the median Hessian eigenvalue on magnetic flux, with error bars indicating the 25th and 75th percentiles.

the loss landscape, Newton's method should converge quickly to the precise minimum. We also used the full wave function vector to compute the gradient and Hessian exactly in this experiment. However, the improvement in the overlap with the exact ground state was only about $10^{-4}$, which is negligible. This indicates that the minimum was successfully found, and local adjustments to the neural network parameters cannot yield improvements significant enough to close the performance gap between cases with $\alpha = 0$ and $\alpha = 0.3$. In conclusion, the results discussed in this section indicate that neither the ruggedness nor the curvature of the loss landscape is the source of the performance issues observed.

## 4.3 Experiments with NQS Modifications

When utilizing NQS, numerous implementation options are available, including choices of loss function, activation function, normalization technique, neural network architecture, and schemes for input and output encoding, etc. In this section, we present the outcomes of experiments involving various such modifications. The results are summarized in Table 1.

### 4.3.1 Complex weights

Given that the ground states of non-stoquastic Hamiltonians inherently involve complex numbers, employing a neural network with complex weights might more efficiently encapsulate their structure. We experimented with complex weights as opposed to real ones. This necessitated rederivation of the gradient expression in Eq. 10 without assuming that the parameters

$\theta$ are real. Additionally, we modified the neural network's activation function so that the GELU function acts independently on the real and imaginary parts. As indicated in Table 1, the adoption of complex weights results in only a marginal improvement in performance. Furthermore, this slight enhancement is likely attributable to the fact that complex weights offer twice as many degrees of freedom as real weights, rather than a greater efficacy of complex weights for this task.

### 4.3.2 MSE loss

We also experimented with an alternative loss function. Rather than employing the overlap loss as detailed in Eq. 9, we tested the mean squared error (MSE) loss, which is frequently used in regression problems. For its computation, we sampled batches of states in two ways: uniformly and according to the NQS wave function (similarly to how it's done with overlap loss). Sampling in accordance with the wave function yielded considerably better results. However, as indicated in Table 1, adopting this loss function resulted in significantly inferior performance compared to the baseline scenario utilizing overlap loss.

### 4.3.3 Input encoding

Different input encoding techniques can significantly affect the expressivity of a neural network because certain input representations can simplify the mapping structure between input and output. In addition to our baseline encoding described in Sec. 3.2, we explored three alternative input encoding schemes. The first method involves applying a two-dimensional Fourier transform to the original input before feeding it into the neural network. The second method utilizes a learnable encoding where all $2 \times 2$ lattice segments undergo a linear transformation into one dimensional vectors with 4 elements, which are then concatenated and used as the neural network input. This scheme is recognized in the deep learning literature as patches encoding [63]. The third method employs learnable embeddings, a technique commonly used in natural language processing. Here, both the lattice position and the number of particles at each site are encoded as learnable vectors with 2 elements, summed, and then concatenated into a single vector for the neural network input. As indicated in Table 1, all three of these input encoding schemes yield performance that is either worse than or similar to our baseline approach, with patches encoding showing the best results among the three methods discussed here.

### 4.3.4 Curriculum learning

For certain challenging machine learning problems, it can be advantageous to train neural networks by beginning with a simpler task and gradually making the task more difficult until it aligns with the ultimate goal. This strategy, known as curriculum learning, often surpasses the effectiveness of training directly on the final task [64]. We attempted to implement this concept by starting with the simpler case of $\alpha = 0$ and incrementally increasing $\alpha$ to 0.3 during the training process. However, as shown in Table 1, this method did not enhance the final outcome and even resulted in slightly inferior performance compared to directly learning the ground state at $\alpha = 0.3$. While continuously increasing $\alpha$, a phase transition occurs at certain values of $\alpha$ that causes an abrupt change in the ground state, leading to a sudden increase in NQS overlap deviation. In such cases, previously learned representations may become irrelevant and thus do not provide a significantly better starting point compared to random initialization. In fact, it appears that in such situations, NQS can become trapped in worse local minima, which could explain the decreased performance compared to training from scratch.

Table 1: Comparison of overlap deviations achieved with various NQS modifications at $\alpha = 0$ and $\alpha = 0.3$. The 'Baseline' row refers to the configuration used in earlier sections.

| Modification | $d(\psi_{\mathrm{NQS}}, \psi_{\mathrm{ED}})$ **at** $\alpha = 0$ | $d(\psi_{\mathrm{NQS}}, \psi_{\mathrm{ED}})$ **at** $\alpha = 0.3$ |
|---|---|---|
| Baseline | $6.95 \cdot 10^{-6}$ | $1.23 \cdot 10^{-2}$ |
| Complex parameters | $5.84 \cdot 10^{-6}$ | $6.51 \cdot 10^{-3}$ |
| MSE loss | $2.72 \cdot 10^{-5}$ | $6.28 \cdot 10^{-2}$ |
| Fourier transformed input | $1.22 \cdot 10^{-5}$ | $1.66 \cdot 10^{-2}$ |
| Patches encoding | $7.80 \cdot 10^{-6}$ | $1.37 \cdot 10^{-2}$ |
| Learnable embeddings | $1.92 \cdot 10^{-5}$ | $1.85 \cdot 10^{-2}$ |
| Float64 | $7.01 \cdot 10^{-6}$ | $1.15 \cdot 10^{-2}$ |
| Curriculum learning | - | $2.14 \cdot 10^{-2}$ |

### 4.3.5 Numerical precision

In all our computations involving neural networks, we used 32-bit floating point numbers, a common practice in deep neural network research due to faster computation on GPUs and reduced memory usage compared to 64-bit floating numbers. However, this could lead to greater numerical instability, potentially affecting performance, especially if accurately finding the ground state in strong magnetic fields requires higher precision. To assess this possibility, we also tried using only 64-bit floating point numbers. As indicated in Table 1, the results were practically the same as those obtained with 32-bit floating numbers. The lack of improvement suggests that numerical precision is likely not a contributing factor to performance issues.

## 5 Conclusion

In this work, we demonstrated that the HBH model of strongly-interacting bosons under a magnetic field exemplifies a system whose ground states are challenging to approximate using unspecialized NQS architectures like MLP or CNN. The energy error increases by up to three orders of magnitude with increasing magnetic flux, and this increase is consistent regardless of variations in the physical model, neural network architecture, or optimization procedure. This universality suggests a significant challenge which might require developing novel architectures or optimization methods that are specifically adapted for this system. To uncover the root cause of the difficulty, we conducted numerical experiments and analyzed potential explanations related to the statistics of wave function elements, complex phase structure, quantum entanglement, fractional quantum Hall effect, and the variational loss landscape. We also explored whether the issue could be mitigated through various minor modifications to NQS and its optimization procedures. Despite extensive experimentation and analysis, the exact cause remains elusive. This makes HBH model particularly interesting as it potentially represents a new type of challenge, different from Heizenberg $J_1$-$J_2$ or Newman-Moore models which were identified in earlier works. It appears that ground states in high magnetic flux regimes present a structure that is hard to exploit for neural networks. However, quantifying this issue is difficult without a robust theory of neural network expressivity. It remains uncertain whether this challenge can be addressed through some modifications or if it represents an example of a more fundamental limitation.

Based on our findings, we propose the model used in this study as an effective testing ground for NQS, which could complement the commonly analyzed Heisenberg $J_1$ - $J_2$ model. HBH model is challenging enough to be interesting even at small sizes, which facilitates rapid experimentation and detailed analysis. Additionally, its complexity can be easily increased, for example, by removing the hard-core boson constraint or introducing not only on-site interactions but also between neighboring sites. Our results also suggest that the difficulties associated with NQS cannot be resolved through simple means, highlighting the complexity and richness of the problem.

## Acknowledgments

The authors express their gratitude to Julius Ruseckas for the thought-provoking discussions.

**Funding information** This research has been carried out in the framework of the "Universities' Excellence Initiative" programme by the Ministry of Education, Science and Sports of the Republic of Lithuania under the agreement with the Research Council of Lithuania (project No. S-A-UEI-23-6).

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
