# Peer review of "Universal Performance Gap of Neural Quantum States Applied to the Hofstadter-Bose-Hubbard Model"

_SciPost Physics, doi:SciPost Phys. 18, 011 (2025)_

## Round 1 · Referee Report · Anonymous (Referee 1) · 2024-6-28

Report

The manuscript is concerned with questions related to the training of Neural Quantum States (NQS), a recent numerical approach for approximating ground states of quantum many-body Hamiltonians. The authors focus on the Hofstadter-Bose-Hubbard (HBH) model, a non-interacting boson system on a two-dimensional square lattice with a perpendicular magnetic field. Using multi-layer perceptron and convolutional neural network architectures, the authors perform numerical experiments in three different setups (Variational Monte Carlo optimization of the energy, supervised imaginary time evolution and supervised learning of exact solution) to consistently show a drop in NQS performance in the strong magnetic flux regime. To uncover this drop in performance, the authors analyzed possible explanations related to the statistics of the wave function elements, the complex phase structure, quantum entanglement, the fractional quantum Hall effect, and the variational loss landscape. However, the precise reason for this degradation remain elusive.

Overall, the paper is well written and the results are clearly presented. However, I don't think that the paper fulfill the journal expectation indicated from the authors, namely, to `Open a new pathway in an existing or a new research direction, with clear potential for multi-pronged follow-up work'. Specifically, I question the choice of the HBH model as a particularly effective testing ground for NQS. With the architectures and optimization protocols outlined in the paper, similar conclusions could likely be reached for many other physical models. For instance, similar results to those presented in Fig. 1 (left panel) of the manuscript could be achieved by optimizing a multi-layer perceptron within the Variational Monte Carlo framework on the J1-J2 Heisenberg model on a 4x4 square lattice. Using Stochastic Reconfiguration and a relatively small batch size (e.g., 512, as considered in the present work), the final relative error would worsen by orders of magnitude when increasing the frustration ratio to J2/J1. However, it is incorrect to draw conclusions such as "there are fundamental limitations of NQS in describing the physics of the 2D J1−J2 Heisenberg model." On the contrary, NQS-based architectures have achieved state-of-the-art ground state energies for this system, surpassing other traditional methods (see for example arXiv:2302.01941, Phys. Rev. X 11, 031034, arXiv:2310.05715). Furthermore, also for bosonic models, NQS can achieve state-of-the-art energies on the 2D Bose-Hubbard model (see https://arxiv.org/pdf/2404.07869). The latter is an interacting model and consequently more complicated than the one described in the paper under review.

Summarizing, while the manuscript contains valid and well-presented calculations, the conclusion that NQS are unsuitable for studying the HBH model is not appropriate. The authors should revise the discussion of their results in the context of their particular setup, avoiding generalizations about the limitations NQS.

As a minor comment, in Fig. 1 (left panel), the y-axis label currently shows E_NQS - E_ED. It would be more informative to report the relative error, abs((E_NQS - E_ED)/E_ED), instead.

Recommendation

Ask for major revision

  • validity: low
  • significance: low
  • originality: ok
  • clarity: high
  • formatting: excellent
  • grammar: excellent

Author:  Eimantas Ledinauskas  on 2024-07-08  [id 4607]

(in reply to Report 1 on 2024-06-28)
Category:
reply to objection
correction

Thank you for your comments. We acknowledge that our text in the introduction and conclusion sections could be improved to more clearly reflect our intended claims, and we will gladly revise it. Below are our replies to some specific statements.

The referee writes:

authors focus on the Hofstadter-Bose-Hubbard (HBH) model, a non-interacting boson system

Our response: Let us stress that the HBH model describes a system of interacting bosons in the limit where interactions are particularly strong. Although the interactions are not directly visible in the Hamiltonian, they are implemented by means of the hard-core constraint. In particular, a recent work [SciPost Phys 12, 095 (2022)] has demonstrated that the model is suitable to describe the fractional quantum Hall regime in lattice systems.

The referee writes:

Specifically, I question the choice of the HBH model as a particularly effective testing ground for NQS. With the architectures and optimization protocols outlined in the paper, similar conclusions could likely be reached for many other physical models. For instance, similar results to those presented in Fig. 1 (left panel) of the manuscript could be achieved by optimizing a multi-layer perceptron within the Variational Monte Carlo framework on the J1-J2 Heisenberg model on a 4x4 square lattice. Using Stochastic Reconfiguration and a relatively small batch size (e.g., 512, as considered in the present work), the final relative error would worsen by orders of magnitude when increasing the frustration ratio to J2/J1.

Our response:
It is true that a drop in NQS performance is also observed in the J1-J2 model, which we mentioned in our introduction. However, it is important to also consider complementary models and gather more such challenging examples to better understand the characteristics of the ground state that lead to this decline. Moreover, our results suggest that the reasons for performance decline in the HBH model might differ from those in the J1-J2 model. For example, unlike [SciPost Phys 10, 147 (2021)], which analyzed the J1-J2 model, we do not find evidence that the decline in the HBH model is due to a rugged variational loss landscape.

The referee writes:

However, it is incorrect to draw conclusions such as "there are fundamental limitations of NQS in describing the physics of the 2D J1−J2 Heisenberg model." On the contrary, NQS-based architectures have achieved state-of-the-art ground state energies for this system, surpassing other traditional methods

Our response: We do not make such a general conclusion (or at least do not intend to). In our conclusions, we specifically write: "It remains uncertain whether this challenge can be addressed through some modifications or if it represents a more fundamental limitation of NQS, potentially requiring problem-specific adjustments.". In this work, we demonstrate a performance decline of several orders of magnitude, which cannot be solved by the various adjustments we described. We think it is important to identify and analyze such cases, even if NQS can achieve better performance than other state-of-the-art methods, because this might lead to realizations on how to improve NQS architectures or optimization procedures. These challenging models are analogous to challenging datasets in the machine learning field, like ImageNet, which historically catalyzed the development of breakthroughs.

We hope that our replies and updated text will change the referee's opinion about the validity and significance of our work.

---

## Round 2 · Referee Report · Anonymous (Referee 2) · 2024-9-4

Report

The authors investigate the performance of Neural Quantum States (NQS) to approximate the ground state of the Hofstadter-Bose-Hubbard model, a non-interacting hardcore boson model on a 2D square lattice coupled to a perpendicular magnetic field. Using numerical experiments for small system sizes with various network and loss function setups, the authors show a consistent drop in NQS performance when approaching the strong magnetic flux regime. Despite exploring different explanations for the reason behind this, the cause for the performance degradation remains elusive.

The paper is well written, and results are clearly presented, but I believe that it is missing a guiding thread and a clear take home message for the reader. In the current form, the observation of the performance degradation is followed by a list of explanations that do not explain it. Outlining the conducted experiments in the main text is valid and important, but since these do not explain the observed phenomena, I think the details should be reported in an appendix/supplementary. Without an explanation for the performance degradation, or a proposal how to mitigate it (even if just by making the network bigger; see below), this seems not to be sufficiently engaging.
The authors state that they restrict the network size to be smaller than the Hilbert space dimension. Previous works on NQS performance [see in particular Phys. Rev. Lett. 131, 036502] have also investigated the number of required network parameters to accurately represent a given ground state. Missing a reason for the performance degradation in this case, I think it would be important to know how to mitigate it. Do I need as many NQS parameters as my hilbert space dimension to represent the ground state accurately? How does the required number of NQS parameters behave as a function of magentic flux? This would at least unveil if the NQS is unable to learn an efficient compression of the wave function at strong magnetic flux or if the observed performance drop has more to do with the optimization scheme(s).

In conclusion, the paper contains valid and well presented calculations. An important missing piece is a reason for the performance drop, or at least a discussion how to mitigate the effect by modifying e.g. network size. Since reporting on drawbacks and shortcomings of emerging methods such as NQS is important (and dearly missing in the literature), I still think the manuscript should be published with appropriate modifications.

Recommendation

Ask for major revision

  • validity: ok
  • significance: low
  • originality: ok
  • clarity: high
  • formatting: excellent
  • grammar: excellent

Author:  Eimantas Ledinauskas  on 2024-10-14  [id 4865]

(in reply to Report 1 on 2024-09-04)
Category:
remark
correction

Thank you for your comments and suggestions.

We agree that it is somewhat disappointing that we did not identify the exact cause or solution for the performance decline. However, publishing negative results is also important for scientific progress, and we hope these findings will contribute to the search for a solution. The central focus of our paper is to introduce the HBH model as a new, challenging testing ground for NQS methods, which is difficult to solve and relevant to broader physics research.

Regarding your remarks about the number of parameters required to accurately represent the ground state, we find this to be an interesting question. We have added a new Figure 2 and a paragraph discussing this topic in Section 4.1.

In your report, you wrote “Hofstadter-Bose-Hubbard model, a non-interacting hardcore boson model”. Let us stress that the HBH model describes a system of interacting bosons in the limit where interactions are particularly strong. Although the interactions are not directly visible in the Hamiltonian, they are implemented by means of the hard-core constraint. In particular, a recent work [SciPost Phys 12, 095 (2022)] has demonstrated that the model is suitable to describe the fractional quantum Hall regime in lattice systems.

---

## Round 2 · Referee Report · Anonymous (Referee 3) · 2024-10-8

Strengths

The manuscript provides an extensive numerical study on different NQS architectures, hyper-parameters, optimization strategies and potential challenges for applying NQS to the HBH model, and could hence be a useful reference for the community.

Weaknesses

Despite the extensive study of potential challenges, the authors did not succeed in lowering the error in some regimes of the phase diagram. Given that many challenges identified by the authors are still a matter of current research, I think that the manuscript is still a valuable reference for the field.

Report

In their manuscript, the authors provide a detailed study of NQS approaches for the Hofstadter-Bose-Hubbard model. The manuscript is clearly written, provides useful references to previous works in the field as well as detailed information on the calculations that were done. The approaches to identify (and hopefully at some point resolve) the challenges of the NQS approach to represent this system could be a valuable reference for future works in the field. Hence, I recommend this work for publication after revising / answering the following comments and questions.

Remarks: - Fig. 1: The authors mention that their DMRG results were obtained with a bond dimension of 8 (corresponding to a similar number of parameters as used for the NQS), which seems very small. It would be very important to see how the results compare to a MPS calculation with more parameters, requiring e.g. similar computation time as the NQS results. - Maybe Eq. (11) could be moved into Sec. 3.3.3?

Questions: - As far as I understand, both SR and SITE perform imaginary time evolution, but the difference between them is that different distance measures between are applied (Fubini study distance vs. overlap). Is this correct? In any case, it would be helpful for the reader if the differences / similarities would be highlighted in the manuscript. - Concerning Sec. 4.2.1.: The authors point out that there is a difference between the statistics of the wave function elements for different alpha. Is it somewhere mentioned if the respective NQS wave functions were trained with the complete set of samples? If not, would Fig. 2 look better if it was trained with the complete set? - Concerning the analysis of the minima: "neither the ruggedness nor the curvature of the loss landscape is the source of the performance issues observed". I have 2 questions on this comment: 1) Given the relatively high errors in Fig.1, could it be that the analyzed minimum is not the global minimum? 2) How does the minimum develop when more / less parameters are used?

Recommendation

Ask for minor revision

  • validity: good
  • significance: good
  • originality: good
  • clarity: high
  • formatting: excellent
  • grammar: excellent

Author:  Eimantas Ledinauskas  on 2024-10-14  [id 4866]

(in reply to Report 2 on 2024-10-08)
Category:
remark
answer to question

Thank you for your comments and suggestions. Here are our responses to the two remarks, R1 and R2 and the three questions Q1, Q2, and Q3.

[R1] We added data for DMRG with a virtual bond dimension of 24, where the MPS has approximately four times more parameters than the Hilbert space of the model being studied.

[R2] We moved Eq. 11 to Sec. 3.3.3.

[Q1] We added brief comments on the differences between SR and SITE in Sec. 3.3.2 and included a reference for a more detailed discussion.

[Q2] In all cases we trained the neural networks by using the complete set of the basis vectors in the Hilbert space.

[Q3.1] Regarding the loss landscape, there are no guarantees that the found minimum is the global one because of the local optimization scheme. By stating that “neither the ruggedness nor the curvature of the loss landscape is the source of the performance issues observed”, we mean that we do not observe any significant differences between weak and strong magnetic flux regimes when studying the loss landscape.

[Q3.2] Regarding the development of the identified minimum with different parameter counts, we note that changing the parameter count, such as by increasing the neural network width, alters the network architecture and thus changes the loss landscape in non-trivial ways. Therefore, there is no straightforward way to analyze the evolution of a specific minimum. However, Figures 1 and 2 show that the found minimum becomes deeper as the number of parameters increases.

---

## Round 2 · Author Response

We primarily revised the introduction and conclusion sections to more clearly emphasize the relevance of our work. Additionally, we reformulated some conclusions to be less general. We also included citations and comments about another recent preprint article on the HBH model (arXiv:2405.04472).

---

## Round 2 · List of Changes

• revised introduction section
  • revised conclusion section
  • changed the normalization of energy in Figure 1 as suggested by referee 1

---

## Round 3 · Referee Report · Anonymous (Referee 2) · 2024-10-14

Report

The authors have responded convincingly to my comments.
Hence, I now recommend publication of the manuscript in its present form.

Recommendation

Publish (meets expectations and criteria for this Journal)

---

## Round 3 · Referee Report · Anonymous (Referee 3) · 2024-11-18

Report

With the changes made in the last round of revisions the authors have answered all my concerns / requests and I now recommend the publication of this work in SciPost Physics.

Recommendation

Publish (meets expectations and criteria for this Journal)

---

## Round 3 · List of Changes

• Text below Eq. 10: We have added comments and a reference for more details on the differences between the SR and SITE methods, as suggested by Referee 2.

  • Sec. 3.3.3: We moved Eq. 11 to this section, following the suggestion of Referee 2.

  • Section 4.1, Fig. 1: As recommended by Referee 2, we have included data points for DMRG with a larger virtual bond dimension and added corresponding comments in the text discussing this figure.

  • Last paragraph of section 4.1 and Fig. 2: We provided a more detailed analysis of the NQS energy accuracy as a function of the number of parameters and magnetic flux, as suggested by Referee 1.

---

## Editorial Decision

published